# A modular effector with a DNase domain and a marker for T6SS substrates

Biswanath Jana[1,3], Chaya M. Fridman[1,3], Eran Bosis [2] & Dor Salomon [1]

Bacteria deliver toxic effectors via type VI secretion systems (T6SSs) to dominate competitors, but the identity and function of many effectors remain unknown. Here we identify a *Vibrio* antibacterial T6SS effector that contains a previously undescribed, widespread DNase toxin domain that we call PoNe (Polymorphic Nuclease effector). PoNe belongs to a diverse superfamily of PD-(D/E)xK phosphodiesterases, and is associated with several toxin delivery systems including type V, type VI, and type VII. PoNe toxicity is antagonized by cognate immunity proteins (PoNi) containing DUF1911 and DUF1910 domains. In addition to PoNe, the effector contains a domain of unknown function (FIX domain) that is also found N-terminal to known toxin domains and is genetically and functionally linked to T6SS. FIX sequences can be used to identify T6SS effector candidates with potentially novel toxin domains. Our findings underline the modular nature of bacterial effectors harboring delivery or marker domains, specific to a secretion system, fused to interchangeable toxins.

[1] Department of Clinical Microbiology and Immunology, Sackler Faculty of Medicine, Tel Aviv University, 6997801 Tel Aviv, Israel. [2] Department of Biotechnology Engineering, ORT Braude College of Engineering, 2161002 Karmiel, Israel. [3]These authors contributed equally: Biswanath Jana, Chaya M. Fridman. Correspondence and requests for materials should be addressed to E.B. (email: bosis@braude.ac.il) or to D.S. (email: dorsalomon@mail.tau.ac.il)

Bacteria are social organisms that constantly interact with neighboring cells. To ensure sufficient nutrients and space, bacteria must be able to outcompete their nonkin neighbors. The type VI secretion system (T6SS) is a widespread mechanism used by Gram-negative bacteria to antagonize competitors[1,2]. The T6SS is a multi-protein machine that is often encoded by large gene clusters[3]. It delivers toxins, called effectors, across the bacterial cell envelope and into neighboring cells[4,5]. The T6SS is composed of three sub-complexes: the membrane complex, the baseplate, and the tail[6]. The membrane complex recruits the baseplate to the cell membrane[7]. Consequently, the baseplate serves as a platform for assembling the tail tube. This secreted tube is composed of stacked hexameric rings of a protein called Hcp; these rings are capped by a spike composed of a VgrG trimer and a PAAR repeat-containing protein that sharpens the tip[1,2,8–10]. Effectors that mediate T6SS activity decorate the secreted tube. Contraction of an outer sheath propels the inner tube, with the effectors, out of the cell[11].

Although the T6SS was originally identified as a virulence mechanism[1,5], it is now commonly acknowledged that T6SSs mainly play a role in interbacterial competition[4]. Indeed, most T6SS effectors that have been identified mediate antibacterial activities[12]. Such antibacterial effectors are paired, as bi-cistronic units, with cognate immunity proteins that inhibit the activity of the effectors and thus protect the bacterium from self- or kin intoxication[13]. To date, various antibacterial effector families have been described. Some effectors were shown to target vital bacterial components such as nucleic acids[14,15], cell walls[4,13], cell membranes[16–18], and the essential dinucleotides NAD(+) and NADP(+)[19,20], whereas others target conserved proteins such as FtsZ, which is involved in bacterial cell division[21]. Noteworthy, various T6SS effectors targeting eukaryotic cell processes and components have also been described[22].

T6SS effectors have been identified mainly by examining genes within or adjacent to T6SS gene clusters[23], their homology to known effectors[13,16], or by the presence of an N-terminal delivery domain[5,10,24,25]. N-terminal domains associated with T6SS effectors can be either T6SS-secreted tail components (e.g., VgrG[5], Hcp[26], or PAAR[10]) or specialized domains such as Rhs[27] and MIX[24]; the latter can serve as a marker for T6SS substrates. Notably, many toxins that have been identified as T6SS effectors do not contain a recognizable delivery domain or signal, suggesting that additional delivery domains remain to be discovered.

*Vibrio parahaemolyticus*, an emerging pathogen found in marine environments[28], harbors two T6SSs[29,30]. T6SS1 mediates antibacterial activity and is predominantly associated with pathogenic isolates[24,30,31], whereas T6SS2 is found in all *V. parahaemolyticus* isolates[29]. We previously characterized T6SS1 in the clinical isolate RIMD 2210633 and identified three antibacterial effectors[24]. Two effectors, VP1388 and VP1415, are encoded within the T6SS1 gene cluster and are common to T6SS1 clusters in *V. parahaemolyticus*. VP1388 contains a MIX domain, whereas VP1415 contains a PAAR domain. The third effector, VPA1263, is a MIX-effector present only in a subset of *V. parahaemolyticus* isolates[32]. Several reports revealed that members of the same bacterial species often harbor different T6SS effector arsenals[33–35]. We recently showed that MIX effectors can account for some of the effector diversity in members of the *Vibrionaceae* family[24,25,32]. However, we found that many T6SS-encoding *Vibrionaceae* strains, including *V. parahaemolyticus* isolates, do not contain multiple MIX effectors[25]. Thus, other mechanisms of effector diversification probably exist in vibrios.

Here, we set out to identify non-MIX effectors that diversify T6SS1 repertoires in *V. parahaemolyticus*. To this end, we focus on *V. parahaemolyticus* 12-297/B, an isolate from Vietnam that causes acute hepatopancreatic necrosis disease in shrimp[31].

*Vibrio parahaemolyticus* 12-297/B does not encode MIX effectors, aside from a VP1388-homolog that is encoded within the T6SS1 gene cluster[31]. Therefore, we hypothesize that it harbors non-MIX effectors that diversify its T6SS1 effector arsenal. Indeed, we find an auxiliary T6SS module that contains a VgrG-encoding gene, which we name *vgrG1b*. We show that the two additional genes found in this module constitute a T6SS1 effector/immunity pair. In addition, we demonstrate that the identified effector contains a novel DNase toxin domain, which we name PoNe (Polymorphic Nuclease effector), that is widespread in various antibacterial toxin secretion systems of both Gram-negative and Gram-positive bacteria. Moreover, we describe PoNi (Polymorphic Nuclease immunity), cognate immunity proteins for these DNase toxins that contain DUF1911 and DUF1910 domains. The newly identified effector further reveals an N-terminal domain, which we name FIX (Found in type sIX effector). We show that FIX is genetically and functionally linked to T6SS, and that it is found in many known and predicted T6SS effectors. We propose that FIX can serve as a marker for T6SS substrates.

## Results

**T6SS1 auxiliary module in *V. parahaemolyticus* 12-297/B.** Effectors found outside of the main T6SS gene clusters in vibrios often belong to auxiliary T6SS modules that encode at least one of the tail tube components: VgrG, Hcp, or PAAR[33,34]. Searching for such auxiliary T6SS modules, we performed TBLASTN analysis for genes encoding VgrG, Hcp, or PAAR proteins in the genome of *V. parahaemolyticus* 12-297/B. Our search revealed a gene encoding an orphan VgrG, B5C30_RS14470. This gene was annotated as a pseudogene, yet examination of the open reading frame revealed an intact gene (located between positions 22692 and 20311 in reference sequence NZ_MYFG01000470.1). This sequence is predicted to encode a VgrG protein homologous to VgrG1 of *V. parahaemolyticus*, with an additional C-terminal extension of 101 amino acids (Supplementary Fig. 1). Thus, we named it VgrG1b. *vgrG1b* appears to be the first in a three-gene operon flanked by a tRNA gene and predicted transposases (Fig. 1a), suggesting that this module is mobile. We found the VgrG1b module in the genomes of other *V. parahaemolyticus* strains isolated around the world (Fig. 1b and Supplementary Table 1).

**New T6SS1 antibacterial effector/immunity pair.** Since VgrG1b is very similar to VgrG1, we predicted that the VgrG1b module plays a role in T6SS1 activity. Before testing this, we first confirmed, using secretion and competition assays, that like the previously characterized *V. parahaemolyticus* POR1 (a RIMD 2210633 derivative) T6SS1[24,30], *V. parahaemolyticus* 12-297/B T6SS1 is functional and mediates antibacterial activity under warm marine-like conditions in the presence of surface-sensing activation (Supplementary Fig. 2).

Often, *vgrG*-containing modules encode downstream effectors. Therefore, we hypothesized that the two genes found downstream of *vgrG1b*, namely, B5C30_RS14465 (encoding WP_029857615.1) and B5C30_RS14460 (encoding WP_024703575.1) (hereafter named *v12_14465* and *v12_14460*, respectively), encode an effector/immunity (E/I) pair of T6SS1 (Fig. 1a). Consistent with this premise, deletion of the genes encoding the putative E/I pair (Δ*v12_14465-0*) resulted in the loss of immunity against T6SS1-medited self-intoxication. However, immunity was restored upon exogenous expression of V12_14460, thus confirming its role as an immunity protein (Fig. 1c). Self-intoxication, but not overall T6SS1-mediated antibacterial activity against *Escherichia coli* prey, was dependent on *v12_14465*, confirming that it is a *bona*

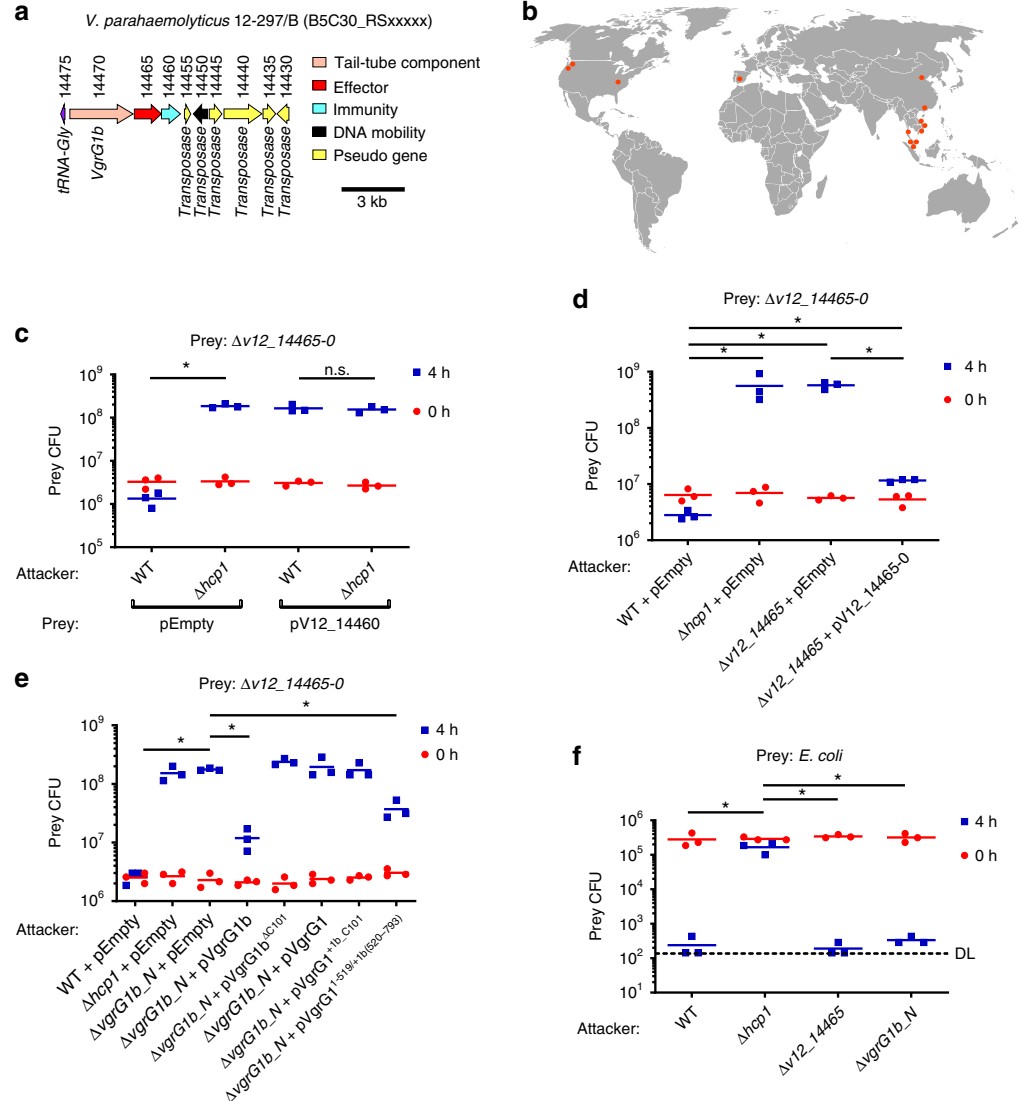

**Fig. 1** An auxiliary T6SS1 module in *V. parahaemolyticus* 12-297/B. **a** T6SS1 VgrG1b auxiliary module. Genes are represented by arrows indicating the direction of translation. Locus tags (B5C30_RSxxxx) are shown above. **b** Locations in which the *V. parahaemolyticus* isolates containing the VgrG1b module have been isolated are denoted by red circles. Viability counts of *V. parahaemolyticus* 12-297/B (**c–e**) or *E. coli* (**f**) prey before (0 h) and after (4 h) co-incubation with the indicated *V. parahaemolyticus* 12-297/B attackers on media containing 3% NaCl at 30 °C. L-Arabinose was added (0.1% in **c**, 0.01% in **e**) to induce expression from the Pbad promoter. Δ*hcp1* is used as T6SS1⁻ control. Asterisks denote statistical significance between samples at the 4 h timepoint by unpaired, two-tailed Student's *t* test (*$P < 0.05$); n.s. no significant difference, WT wild-type, DL detection limit. Source data are provided as a source data file

*fide* T6SS1 antibacterial effector (Fig. 1d, f). Notably, exogenous over-expression of V12_14465 alone proved toxic to expressing cells, and we therefore used the *v12_14465-0* E/I bi-cistronic unit (pV12_14465-0) to complement *v12_14465* deletion.

Since the effector V12_14465 is encoded adjacent to *vgrG1b*, we hypothesized that its delivery is dependent on the presence of VgrG1b. Indeed, deletion of *vgrG1b* (Δ*vgrG1b*_N, deletion of nucleotides encoding the first 724 amino acids) hampered V12_14465-mediated self-intoxication, which was comparable to inactivation of T6SS1 by *hcp1* deletion (Fig. 1e). However, the deletion of *vgrG1b* did not affect the overall T6SS1-mediated antibacterial activity against *E. coli* prey, indicating that other T6SS1 effectors were still delivered in its absence (Fig. 1f). Notably, deletions of *hcp1*, *v12_14465*, or *vgrG1b* did not affect *V. parahaemolyticus* 12-297/B growth (Supplementary Fig. 3).

We further hypothesized that the 101 residues C-terminal extension, found in VgrG1b but absent in VgrG1 (Supplementary

Fig. 1), plays a role in the delivery of the downstream effector. Indeed, this C-terminal extension was required for effector delivery since ectopic expression of VgrG1b in which we removed the 101 C-terminal residues (VgrG1b^{ΔC101}) was unable to complement the loss of bacterial killing in the Δ*vgrG1b*_N strain (Fig. 1e). Yet, this C-terminal extension was not sufficient to allow delivery of the effector, as fusing this extension to the T6SS1 cluster-encoded VgrG1 (VgrG1^{+1b_C101}) did not complement the Δ*vgrG1b*_N strain. We noted that considerable differences are also found in the sequences corresponding to residues 520–692 of the two VgrG homologs (Supplementary Fig. 1). Therefore, we tested whether additional information required for effector delivery is found in this region of VgrG1b. As shown in Fig. 1e, swapping residues 520–692 of VgrG1 with residues 520–793 of VgrG1b (VgrG1^{1-519/+1b(520–793)}) resulted in partial complementation of the Δ*vgrG1b*_N strain. Taken together, our results indicate that *v12_14465-0* encodes a T6SS1 antibacterial E/I pair

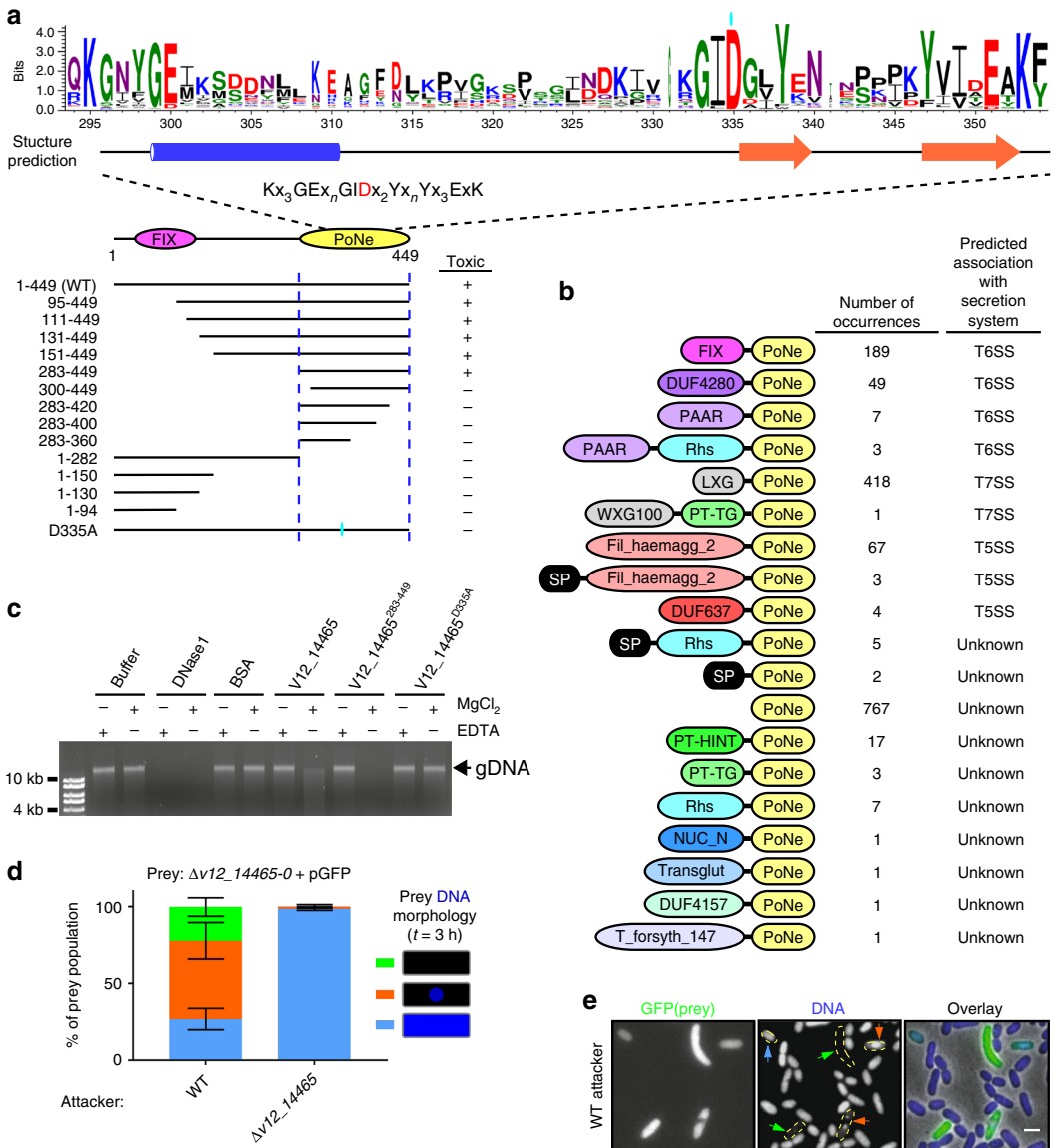

**Fig. 2** Effector V12_14465 contains a new DNase domain. **a** Schematic representation of full-length and truncated V12_14465 forms tested for toxicity upon expression in *E. coli*. The region necessary and sufficient for toxicity is denoted by dashed vertical blue lines (based on the results shown in Supplementary Fig. 5). The region of the toxic C-terminal domain (PoNe) in which a conserved motif was identified (residues 294–354) is illustrated using WebLogo 3. Secondary-structure prediction (Jpred) and the conserved motif are provided. Alpha helix is denoted as blue cylinder, and beta strands as orange arrows. A cyan oval above the WebLogo denotes the catalytic aspartic acid at position 335. **b** Domain architectures of PoNe-containing proteins. SP, signal peptide. **c** In vitro DNase activity assay. Purified *E. coli* genomic DNA (gDNA) was co-incubated with buffer or with the indicated purified proteins in the presence (+) or absence (−) of $Mg^{2+}$ or EDTA at 37 °C for 30 min. The integrity of gDNA was visualized on 1% agarose gel. **d** Quantification of *V. parahaemolyticus* 12-297/B Δ*v12_14465-0* prey cells, constitutively expressing GFP from a plasmid (pGFP), that present DNA morphologies: DNA across entire cell area (cyan), DNA nucleoid or foci (orange), or no DNA (green). DNA was detected using Hoechst dye fluorescence signal after 3 h of co-incubating prey with the indicated attacker strains. Results shown are average ± S.D. of three independent biological replicates. In each experiment, 100 prey cells were evaluated per treatment. **e** Sample images of prey cells described in **d** after 3 h of co-incubation with WT attackers. Dashed yellow shapes in DNA channel encircle prey cells detected in the GFP channel. Colored arrows denote prey cells presenting each of the three DNA morphologies described in **d**. Bar = 2 μm. WT wild-type. Source data are provided as a source data file

whose delivery is dependent on VgrG1b. Since all tested VgrG forms were stably expressed in *V. parahaemolyticus* (Supplementary Fig. 4), we also conclude that the C-terminal end of VgrG1b contains information required and sufficient for delivery of its downstream effector.

**V12_14465 belongs to a widespread family of new DNase toxins**. Analysis of V12_14465 revealed no sequence homology to known toxin domains. Therefore, we set out to decipher the

antibacterial mechanism of action and target of this effector. Exogenous expression of V12_14465 in *E. coli* was detrimental (Fig. 2a and Supplementary Fig. 5). Truncation analysis indicated that the V12_14465 C-terminal end, encompassing residues 283–449, was necessary and sufficient for the observed toxicity in *E. coli* (Fig. 2a and Supplementary Fig. 5). We named this toxic domain PoNe (as explained below). Using position-specific iterated BLAST (PSI-BLAST), we identified 1546 proteins containing a PoNe domain (Supplementary Data 1); these

proteins were widely distributed in the genomes of both Gram-negative and Gram-positive bacteria (Supplementary Fig. 6). Many PoNe domains were fused to N-terminal polymorphic toxin delivery domains of T6SS (e.g., PAAR[10] and DUF4280[36]), type VII secretion system (T7SS; e.g., LXG[37] and WXG100[38]), and type V secretion system (T5SS) contact-dependent inhibition (e.g., DUF637 and Fil_haemagg_2[39,40]); they were even found fused to predicted Sec pathway (T2SS) signal peptide (Fig. 2b and Supplementary Data 1). These results suggest that V12_14465 belongs to a widespread family of antibacterial effectors associated with various protein secretion systems.

When we analyzed the multiple sequence alignment of the identified PoNe family members, we discovered a conserved motif: $Kx_3GEx_nGIDx_2Yx_nYx_3ExK$ (Fig. 2a). We noted that the PoNe conserved motif and its position within the predicted secondary structure resembled the conserved motif of members of the diverse PD-(D/E)xK phosphodiesterase superfamily, many of which target DNA[41–43]. Therefore, we hypothesized that PoNe constitutes a new family of phosphodiesterases within this superfamily, and that V12_14465 is a nuclease that targets DNA. Indeed, purified V12_14465 or its PoNe domain ($V12\_14465^{283–449}$) exhibited magnesium-dependent DNase activity in vitro (Fig. 2c). We reasoned that if V12_14465 acts as a DNase inside bacterial cells, then its activity should activate the SOS response, a mechanism used by bacteria to cope with DNA damage. As predicted, expression of V12_14465, or its PoNe domain, induced the expression of a fluorescent SOS response reporter in E. coli (Supplementary Fig. 7), consistent with PoNe inducing DNA damage. Elevated SOS signals were also detected under repressing conditions, possibly due to leaky expression which was also evident by visible toxicity (Supplementary Fig. 5a). Moreover, substitution of the conserved V12_14465 aspartic acid 335 for alanine (D335A), which is expected to play a role in the catalytic activity of the PD-(D/E)xK phosphodiesterase superfamily[41], abolished bacterial toxicity (Fig. 2a and Supplementary Fig. 5), in vitro DNase activity (Fig. 2c), and SOS response activation (Supplementary Fig. 7). Noteworthy, when we extracted total RNA from E. coli cells expressing V12_14465 we did not observe differences compared with control cells, suggesting that PoNe targets DNA rather than RNA (Supplementary Fig. 8).

To further test the hypothesis that PoNe-containing V12_14465 functions as a DNase during bacterial competition, we used fluorescence microscopy. We visualized the DNA in prey cells lacking the v12_14465-0 E/I pair (Δv12_14465-0) when competed against WT V. parahaemolyticus 12-297/B attacker or against a mutant in which the v12_14465 effector was deleted (Δv12_14465). As shown in Fig. 2d, e, prey cells in which DNA was no longer visible were detected after 3 h incubation with the WT attacker, but not when prey were competed against an attacker lacking the effector v12_14465. Taken together, these results demonstrate that V12_14465 is a bona fide antibacterial DNase toxin.

Next, we aimed to determine whether other members of the PoNe family also possess DNase activity. To this end, we expressed in E. coli the Bacillus cereus ATCC 14579 protein BC3021 (NP_832767.1). This protein harbors a C-terminal PoNe domain fused to an N-terminal region containing a T7SS LXG delivery domain. As we observed for V12_14465, the PoNe-containing B. cereus BC3021 was detrimental to E. coli, activated the SOS response, and induced DNA degradation in vivo (Supplementary Fig. 9). Taking the above results together, we conclude that PoNe constitutes a new family of DNases that is widespread in antibacterial toxins associated with various secretion systems.

**DUF1911-containing proteins provide immunity against PoNe.** V12_14460, the immunity protein encoded downstream of effector v12_14465, contains a domain of unknown function 1911 (DUF1911). Out of 1546 PoNe-encoding genes, 1389 (89.84%) are found immediately upstream of a gene that encodes a protein containing DUF1911. DUF1911 is often also accompanied by a DUF1910 domain (Supplementary Data 1). This observation implied that DUF1911- and DUF1911-DUF1910-containing proteins are PoNe immunity proteins (hereafter referred to as PoNi). We investigated whether the DUF1911-containing V12_14460, which was required for immunity against V12_14465-mediated self-intoxication in V. parahaemolyticus (Fig. 1c), was sufficient to provide immunity against V12_14465 expressed in E. coli. As expected, expression of V12_14460 ($PoNi^{Vp}$) specifically abrogated the toxicity mediated by the V12_14465 PoNe domain (residues 283–449; $V12\_14465^{PoNe}$) in E. coli, but did not provide immunity against VPA1263, a T6SS1 effector from V. parahaemolyticus RIMD 2210633 with a predicted colicin DNase domain[24] (Fig. 3a). Next, we examined whether $PoNi^{Vp}$ provided immunity by direct interaction with the V12_14465 PoNe domain. Indeed, direct interaction between $PoNi^{Vp}$ and PoNe domain-containing variants of V12_14465, but not between $PoNi^{Vp}$ and the N-terminal portion of V12_14465 (residues 1–282) was apparent in both pull-down and bacterial two-hybrid (BACTH) assays (Fig. 3b, c). Accordingly, the DUF1911-containing PoNi BC3020 (accession number: NP_832766.1; hereafter referred to as $PoNi^{Bc}$), encoded downstream of the B. cereus PoNe-containing toxin BC3021, was sufficient to rescue E. coli expressing the B. cereus PoNe domain ($BC3021^{PoNe}$) or the full-length BC3021 protein (Supplementary Fig. 9b). Notably, PoNi proteins appear to provide immunity specifically against their adjacently-encoded PoNe-containing effector, since the Bacillus and Vibrio PoNi were unable to cross-protect against a noncognate PoNe toxin (Supplementary Fig. 9e). These results indicate that DUF1911 domains, which are often accompanied by DUF1910, provide immunity against PoNe-containing DNase toxins, most likely by directly interacting with them.

**The N-terminus of V12_14465 contains a new T6SS marker.** Next, we set out to characterize the N-terminal portion of effector V12_14465. Since PoNe is often fused to N-terminal effector delivery domains (Fig. 2b), we reasoned that the N-terminal part of V12_14465 contains a yet-unknown domain that is required for its T6SS-mediated delivery into recipient cells. Consistent with this premise, V12_14465 N-terminal truncations were unable to complement a deletion in v12_14465 during self-competition against a Δv12_14465-0 prey (Fig. 4a), even though these truncations were still functional toxins as evident by their toxicity upon expression in E. coli (Supplementary Fig. 5). This indicated that the N-terminal part of V12_14465 is required for its T6SS1-mediated delivery. Using PSI-BLAST, we identified 2881 proteins containing a region homologous to the V12_14465 N-terminus. Interestingly, the identified proteins are almost exclusively encoded by Gram-negative bacteria (Supplementary data 2). Multiple sequence alignment of the homologous sequences revealed a possible domain of ~80 amino acids, corresponding to positions 43–121 in V12_14465, containing a conserved motif (Fig. 4b). Since this domain was first identified in T6SS effector V12_14465, we named it FIX (Found in type sIX effector).

Analysis of FIX-containing proteins showed that FIX is often fused to various C-terminal nuclease toxin domains, including PoNe, or to C-terminal extensions of unknown function (Fig. 4c and Supplementary Data 2). Interestingly, FIX is also present in various established T6SS-secreted effectors that have an

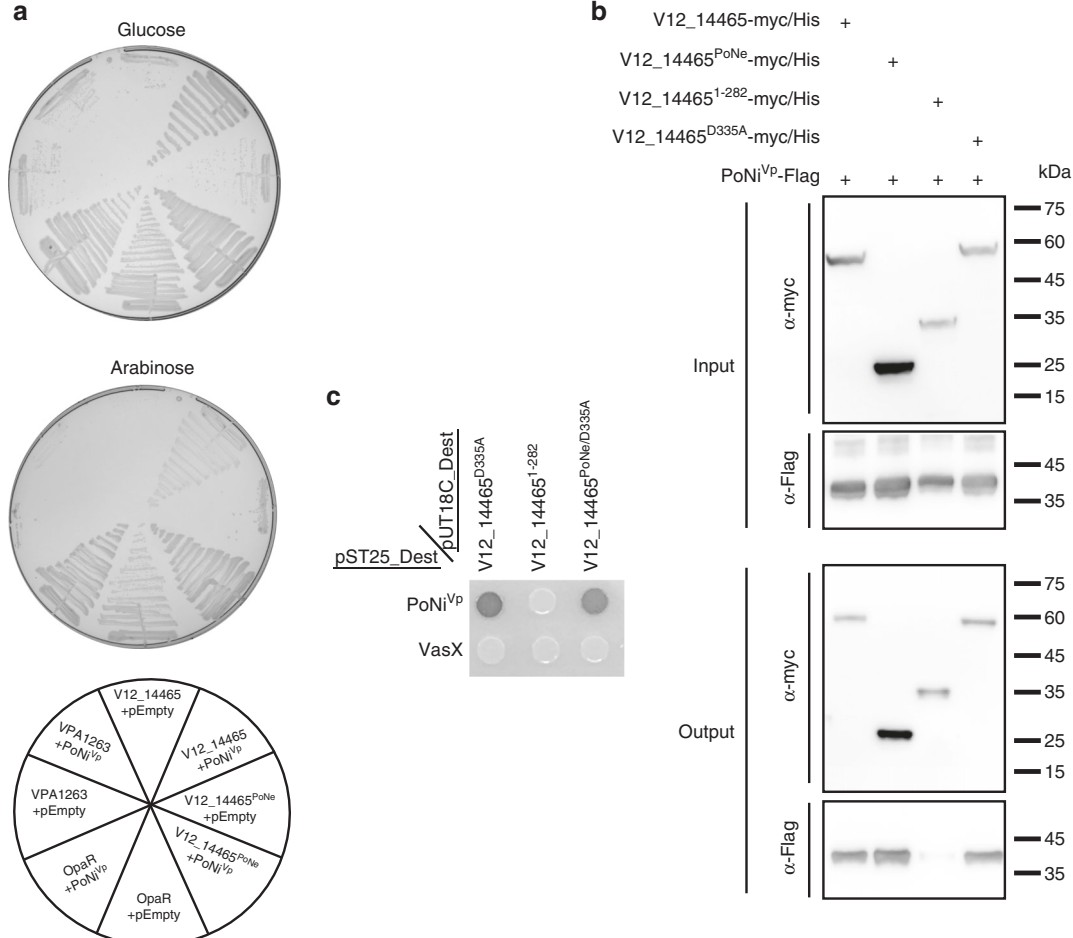

**Fig. 3** V12_14460 provides immunity against PoNe domain. **a** Toxicity of V12_14465 variants expressed in *E. coli* BL21(DE3) from arabinose-inducible expression plasmid with or without V12_14460 (PoNi$^{Vp}$). OpaR, the *V. parahaemolyticus* high cell density quorum sensing master regulator, was used as a nontoxic control. VPA1263, a T6SS1 effector with a predicted colicin DNase domain, was used as a non-PoNe toxic control. **b** Ni-Sepharose resin pull-down of myc/His-tagged V12_14465 forms with PoNi$^{Vp}$-Flag after co-expression in *E. coli* BL21(DE3). **c** Bacterial two-hybrid assay. The indicated proteins were expressed in *E. coli* BTH101 reporter strain fused to the T18 or T25 domain of the *Bordetella* adenylate cyclase. Bacteria were spotted on plates supplemented with the chromogenic substrate X-Gal. A dark-colored colony indicates protein–protein interaction. The *V. cholerae* T6SS effector VasX was used as a negative control. PoNi$^{Vp}$, V12_14460; V12_14465$^{PoNe}$, V12_14465(283–449); V12_14465$^{PoNe/D335A}$, V12_14465(283–449/D335A); pEmpty, empty expression vector. Source data are provided as a source data file

N-terminal VgrG or PAAR domain (e.g., *V. parahaemolyticus* T6SS1 AHH nuclease domain-containing effector VP1415[24]) (Fig. 4c and Supplementary Data 2). The presence of T6SS-related domains such as PAAR and VgrG in FIX-containing proteins, but not of domains associated with other effector secretion systems, suggested that FIX may be specifically linked to T6SS. Indeed, FIX-containing proteins are encoded almost exclusively (>96%) in bacterial genomes that harbor T6SS (Fig. 4d and Supplementary Data 3). In agreement with this observation, we identified genes encoding T6SS components, such as the structural components VgrG[5] and TssL[7], or the adapter DUF4123[44,45], in 70.14% of the genes found directly upstream to FIX (Fig. 4e and Supplementary Data 2). An additional 3.71% of FIX-containing proteins can also be directly linked to T6SS because they contain PAAR or PAAR-like (i.e., DUF4280) domains (Supplementary Data 2). These findings suggested that FIX plays a role in delivery of T6SS effectors. In support of such a role, we found that deletion of the central FIX region encompassing residues 70–89 of the effector V12_14465 (V12_14465$^{\Delta70-89}$) abrogated the effector's ability to mediate toxicity during bacterial competition (Fig. 4f). Noteworthy, V12_14465$^{\Delta70-89}$ retained its toxic activity when exogenously expressed in *E. coli* (Supplementary Fig. 5).

Based on these results, we propose that FIX plays a role in T6SS effector delivery, and that it can serve as a new marker for T6SS-delivered proteins to enable the identification of novel T6SS substrates.

## Discussion

In this work, we identified a VgrG module containing a new T6SS antibacterial effector/immunity pair in *V. parahaemolyticus*. We revealed a new bacterial DNase toxin family, which we named PoNe, that belongs to the PD-(D/E)xK superfamily of phosphodiesterases[42,43]. PoNe is widespread and found in various Gram-negative and Gram-positive bacterial species. Because of the extreme divergence in sequence and structure of PD-(D/E)xK superfamily members, it was impossible to simply identify PoNe by homology searches[42]. Like previously reported toxins belonging to the Tox-REase class[46,47], PoNe possesses the quintessential catalytic D-(D/E)xK residues in secondary-structure positions that are typical of members of this superfamily. However, the broad conserved motif that we identified in PoNe, which surrounds these core residues, is different from motifs found in Tox-REase toxins, indicating that PoNe represents a new and distinct toxin family. This conclusion is further supported by the

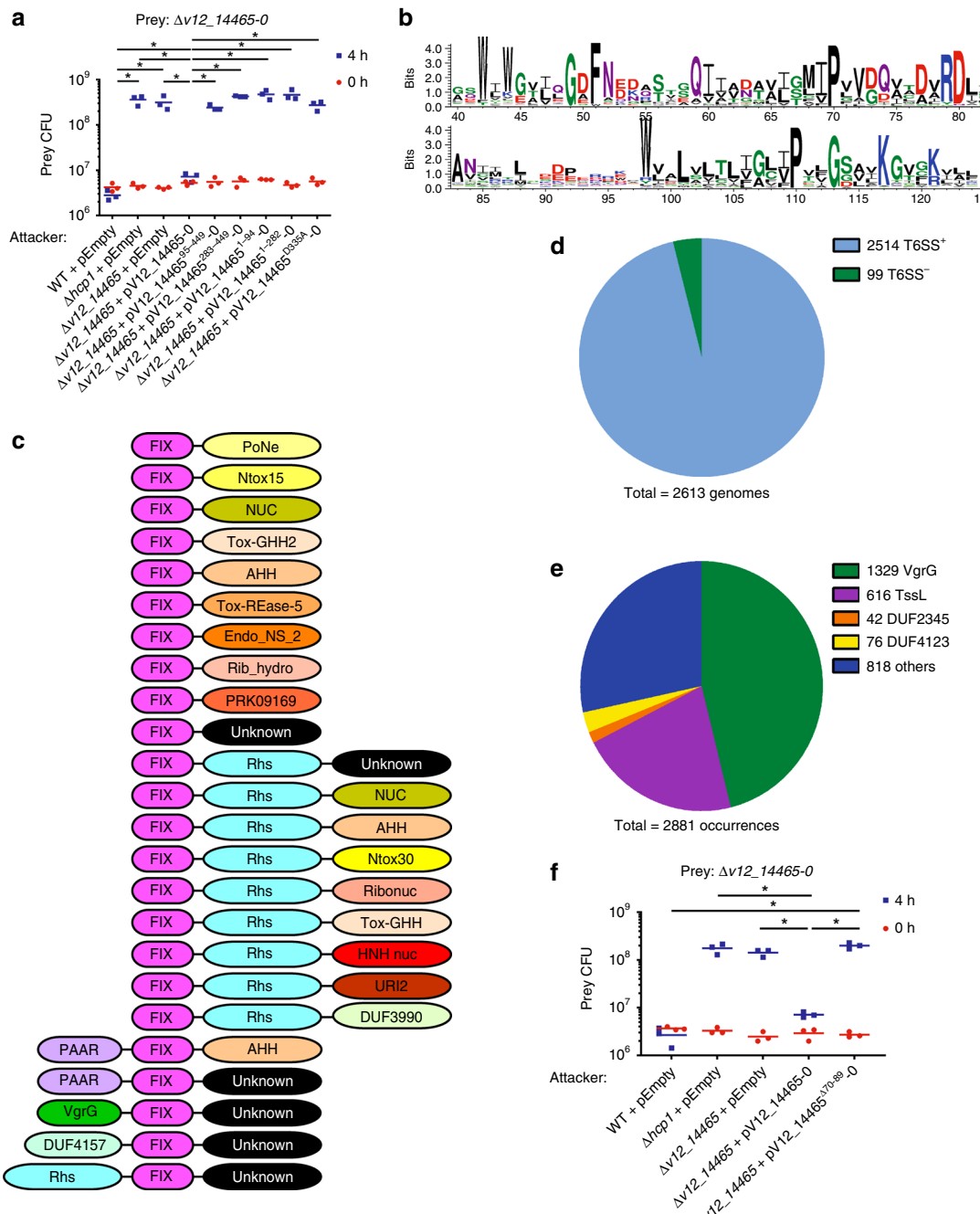

**Fig. 4** V12_14465 contains a marker for T6SS substrates. **a** Viability counts of *V. parahaemolyticus* 12-297/B prey before (0 h) and after (4 h) co-incubation with the indicated *V. parahaemolyticus* 12-297/B attackers at 30 °C on media containing 3% NaCl. **b** Conservation pattern found in FIX homologs. The region shown corresponds to positions 40–125 in V12_14465. The image was produced using WebLogo 3. **c** Representative domain architectures of FIX-containing proteins. **d** Pie chart of FIX-containing bacterial genomes that encode a T6SS (encoding at least 9 T6SS core components). **e** Pie chart of domains encoded by genes found immediately upstream of FIX. The number of occurrences for each domain is listed next to its name. **f** Viability counts of *V. parahaemolyticus* 12-297/B prey before (0 h) and after (4 h) co-incubation with the indicated *V. parahaemolyticus* 12-297/B attackers at 30 °C on media containing 3% NaCl. Asterisks denote the statistical significance between samples at the 4 h timepoint by unpaired, two-tailed Student's *t* test (*$P < 0.05$); WT wild-type. Source data are provided as a source data file

wide distribution of PoNe domains in bacterial species, and by its association with various effector secretion systems, such as T5SS, T6SS, and T7SS. Importantly, to the best of our knowledge, PoNe is the first toxin family of PD-(D/E)xK phosphodiesterases that has been experimentally demonstrated to have DNase activity.

We further showed that proteins containing DUF1911 and DUF1910, which we named PoNi, function as specific immunity proteins that bind and antagonize PoNe. Notably, DUF1911 and DUF1910 can be found downstream of proteins in which the PoNe domain was not detected. This can be explained, at least in part, by the presence of PoNe-like toxin domains (which did not pass our conservative threshold for PoNe family members; see the Methods section) that employ PoNi-like immunity proteins. In other instances, PoNi may be an orphan immunity gene, as was previously demonstrated for various T6SS immunity genes[13,48].

The modular nature of PoNe-containing toxins employing delivery domains of diverse secretion systems prompted us to investigate the N-terminal end of effector V12_14465. This led us to identify a region of ~80 residues that we named FIX. Like the previously reported T6SS marker MIX, FIX is genetically linked to T6SS. FIX is encoded almost exclusively by bacteria that harbor T6SS (>96% of genomes encoding FIX also harbor T6SS, whereas only ~25% of Gram-negative bacterial genomes are expected to encode T6SS[49]). FIX-encoded genes are often neighbors of T6SS component-encoding genes such as VgrG or TssL, and FIX is found fused to known T6SS delivery domains such as PAAR or VgrG. Moreover, FIX is found N-terminal to diverse known toxin domains. Interestingly, most of these toxin domains, like PoNe, are predicted to target nucleic acids. These findings led us to propose that FIX can be used as a new marker for T6SS substrates. Therefore, it is likely that many, if not all, FIX-containing proteins are T6SS effectors. Importantly, aside from MIX, FIX is the only currently available marker for T6SS-secreted proteins that is not a T6SS structural component (i.e., Hcp, VgrG, and PAAR). Future studies of FIX proteins containing C-terminal domains of unknown function may lead to the discovery of novel toxin domains.

We noted that the T6SS markers, MIX[24] and FIX, appear to be mutually exclusive because we did not find proteins containing both of them. This observation implies different or competing mechanisms of T6SS association or roles in the delivery process for these two markers. Future work will determine whether FIX plays either a structural or functional role in T6SS-mediated delivery, such as binding to T6SS-secreted structural components. Although we showed that FIX is required for delivery of a T6SS effector, we have not been able to determine whether FIX is also a *bona fide* T6SS secretion domain sufficient for T6SS-mediated delivery, or whether it can mediate binding directly to a T6SS tail component. Thus, elucidating FIX's exact function will be addressed in future studies.

In conclusion, we identified a new DNase toxin family and its cognate immunity domains, and we also revealed a new marker for T6SS substrates. Our findings illustrate the modular nature of secreted bacterial effectors, and demonstrate the potential to use such modular domains for discovering new toxins.

## Methods

**Strains and media**. *Vibrio parahaemolyticus* isolates 12-297/B[31] (gift from Donald Lightner at the University of Arizona), RIMD 2210633 (gift from Kim Orth at the University of Texas Southwestern Medical Center), POR1[50] (RIMD 2210633 Δ*tdhAS* derivative), and their derivatives were routinely grown in Marine Lysogeny broth (MLB; LB media supplemented with NaCl to a final concentration of 3% wt/vol) or on marine minimal media (MMM) agar plates (2% wt/vol NaCl, 0.4% wt/vol galactose, 5 mM MgSO₄, 5 mM K₂SO₄, 77 mM K₂HPO₄, 35 mM KH₂PO4, 20 mM NH₄Cl, 1.5% wt/vol agar) at 30 °C.

*Escherichia coli* strain DH5α (λ *pir*) (gift from Eric V. Stabb at the University of Georgia) was used for plasmid maintenance and amplification. *E. coli* strain DH5α was used as prey in bacterial competition assays. *E. coli* strain BL21(DE3) was used for protein expression and for toxicity assays. *E. coli* strain BTH101 (from BACTH system kit) was used for BACTH experiments. *E. coli* were routinely grown in 2xYT media (1.6% wt/vol tryptone, 1% wt/vol yeast extract, 0.5% wt/vol NaCl), or on LB (1% wt/vol NaCl) agar plates at 37 °C. When necessary, media were supplemented with 10 µg/mL chloramphenicol, 30 µg/mL (for *E. coli*) or 250 µg/mL (for *V. parahaemolyticus*) kanamycin, 100 µg/mL ampicillin, or 100 µg/mL spectinomycin. To induce the expression of genes from a plasmid, 0.01, 0.05, or 0.1% (wt/vol) L-arabinose was included in the media.

**Plasmid construction**. For arabinose-inducible expression in bacteria, DNA fragments corresponding to the full-length or truncated forms of B5C30_RS14465 (V12_14465) or BC3021 were amplified from *V. parahaemolyticus* 12-297/B or *B. cereus* ATCC 14579 genomic DNA, respectively. PCR fragments were inserted into the multiple cloning site (MCS) of the pBAD/Myc–His vector (Invitrogen) harboring a kanamycin-resistance cassette[30], in-frame with the C-terminal *Myc*-6xHis tag.

For arabinose-inducible expression of immunity genes, the coding sequence of B5C30_RS14460 (V12_14460, PoNi^VP) or BC3020 (PoNi^Bc) was amplified with an

in-frame C-terminal Flag tag from *V. parahaemolyticus* 12-297/B or *B. cereus* ATCC 14579 genomic DNA, respectively. PCR fragments were inserted into the MCS of pBAD33.1 (Addgene).

For complementation of the *v12_14465* deletion strain, the indicated *v12_14465* variants were cloned 5′ of the *v12_14460* coding sequence, and 3′ of the predicted native promoter region of the *vgrG1b-v12_14465-v12_14460* operon (426 bp upstream of the *vgrG1b*, B5C30_RS14470, start site). L-Arabinose was not added to induce expression from these plasmids in *Vibrio*, since expression was mediated by the *vgrG1b* native promoter. For complementation of *vgrG1b* deletion, the *vgrG1* coding sequence (B5C30_RS15295), the *vgrG1b* coding sequence (found between positions 22692 and 20311 in reference sequence NZ_MYFG01000470.1) or its truncated versions, and fusions of VgrG1 with VgrG1b were cloned into the MCS of pBAD/Myc–His vector (Invitrogen) harboring a kanamycin-resistance cassette, with a C-terminal Flag tag.

For bacterial expression of Tse1 (NP_250535.1), a peptidoglycan-hydrolase T6SS effector[4], its coding sequence was amplified from *Pseudomonas aeruginosa* PAO1 genomic DNA and inserted into pPER5[25] in-frame with an N-terminal PelB signal peptide and a C-terminal *Myc*-6xHis tag. The Gibson assembly method[51] was employed to generate the above plasmids.

For bacterial expression of VPA1263, a T6SS1 effector with predicted colicin-like DNase domain[24], its coding sequence was amplified from *V. parahaemolyticus* RIMD 2210633 genomic DNA and inserted between the SacI and KpnI sites in the MCS of pBAD2[30] in-frame with a C-terminal *Myc*-6xHis tag. Construction of the plasmid for expression of OpaR was previously reported[30].

To generate V12_14465^D335A mutant, site-directed mutagenesis was performed using the above-mentioned pBAD/Myc–His plasmids harboring *v12_14465* as a template to generate pV12_14465^D335A. This construct served as a template for amplification and the subsequent cloning of the D335A variant into subsequently described plasmids.

Plasmids used for the BACTH assay were constructed using the Gateway cloning method. The indicated *v12_14465* forms were amplified from pV12_14465^D335A, whereas the *Vibrio cholerae* effector VasX (NP_232421.1) was amplified from *V. cholerae* N16961 genomic DNA. The genes were introduced into the entry plasmid pENTER-D-TOPO using the pENTER-D-TOPO cloning kit (Invitrogen) following the manufacturer's instructions, followed by transfer to the BACTH destination plasmids pST25_DEST or pUT18c_DEST[52] using the Gateway LR clonase II enzyme mix kit (Invitrogen), according to the manufacturer's instructions.

All constructs were confirmed by DNA sequencing.

**Construction of deletion strains**. For in-frame, genomic deletions of *hcp1* (B5C30_RS15290), *vgrG1b* (B5C30_RS14470), *v12_14465* (B5C30_RS14465), or *v12_14465-0* (B5C30_RS14465 - B5C30_RS14460) in *V. parahaemolyticus* 12-297/B, 1 kb sequences upstream and downstream of each specified gene were cloned into pDM4, a Cm^rOriR6K suicide plasmid[53]. pDM4 constructs were introduced into *V. parahaemolyticus* 12-297/B via tri-parental mating using *E. coli* DH5α (λ pir) and a conjugation helper. Trans-conjugants were selected on MMM agar plates containing chloramphenicol (10 µg/mL). The resulting trans-conjugants were grown on MMM agar plates containing sucrose (15% wt/vol) for counter-selection and loss of the SacB-containing pDM4. In-frame, genomic deletion of *V. parahaemolyticus* RIMD 2210633 *hcp1* was performed as described, using pDM4 plasmid described previously[30]. Construction of in-frame, genomic deletion of POR1 *hcp1* was reported previously[30].

**Bacterial competition assay**. Attacker and prey strains were grown overnight in appropriate broth (MLB for *V. parahaemolyticus* and 2xYT for *E. coli*) with the addition of antibiotics when maintenance of plasmids was required. Competition assays were performed as previously described[30]. Briefly, cultures were normalized to OD₆₀₀ = 0.5 and were mixed at a 4:1 ratio (attacker:prey). Triplicates of mixtures were incubated for 4 h at 30 °C on MLB agar plates or on MLB plates containing 0.1 or 0.01% (wt/vol) L-arabinose (when required to induce expression from plasmids). CFU of prey were calculated after they were grown on selective plates at 0 and 4 h. Assays were repeated at least three times with similar results, and the results from representative experiments are shown.

**Protein secretion assay**. Expression and secretion of *V. parahaemolyticus* VgrG1 were determined as previously described[31]. Briefly, overnight cultures of *V. parahaemolyticus* were normalized to OD₆₀₀ = 0.18 in 5 mL of MLB media, and grown for 5 h at 30 °C. Phenamil (20 µM) was added to induce T6SS1[30] (phenamil is an inhibitor of the polar flagella; therefore, it mimics surface-sensing activation[54]). For expression fractions (cells) 1.0 OD₆₀₀ units of cells were collected and resuspended in (2×) tris-glycine SDS sample buffer (Novex, Life Sciences). Supernatants of volumes equivalent to 10 OD₆₀₀ units (media) were filtered (0.22 µm) and pre-cipitated with deoxycholate and trichloroacetic acid[55]. Precipitated proteins were washed twice with ice-cold acetone prior to re-suspension in 20 µL of 10 mM Tris–HCl pH = 8.0, followed by the addition of 20 µL of 2× protein sample buffer. Samples were resolved on TGX stain-free gels (Bio-Rad), transferred onto poly-vinylidene difluoride (PVDF) membranes, and immunoblotted with custom-made anti-VgrG1 antibodies[31] at 1:1000 dilution. Loading of total protein lysates was

visualized by analyzing trihalo compounds' fluorescence of the immunoblot membrane. Protein signals were visualized by enhanced chemiluminescence (ECL). Experiments were repeated at least three times with similar results, and the results from representative experiments are shown.

To verify expression of VgrG1, VgrG1b, and their fusion versions in *V. parahaemolyticus*, cultures were grown as described above, and induced in MLB media containing L-arabinose (0.01%) and appropriate antibiotics for 5 h at 30 °C. A total of 0.5 $OD_{600}$ units of cells were collected and treated as described above. Protein expression was detected in SDS-PAGE using Flag antibodies (DYKDDDDK Tag (D6W5B) Rabbit mAb #14793; Cell Signaling Technology) at 1:1000 dilution.

**Bacterial toxicity assay.** To assess the toxicity of antibacterial effectors, *E. coli* BL21(DE3) were transformed with the indicated pBAD-based L-arabinose-inducible expression vectors. *E. coli* transformants were streaked onto either repressing (containing 0.2% wt/vol glucose) or inducing (containing 0.05% wt/vol L-arabinose) LB agar plates containing kanamycin (30 μg/mL). To assess the protection conferred by immunity proteins, pBAD33.1 L-arabinose-inducible expression vectors carrying predicted immunity genes were co-transformed with the plasmids mentioned above into *E. coli* BL21(DE3). Transformants were streaked onto either repressing or inducing LB agar plates containing kanamycin (30 μg/mL) and chloramphenicol (10 μg/mL). Bacterial growth was assessed after overnight incubation at 37 °C.

**Vibrio growth assay.** Overnight-grown cultures of *V. parahaemolyticus* were normalized to an $OD_{600}$ of 0.01 in MLB media and transferred to 96-well plates (200 μL per well; *n* = 10). Cultures were grown at 30 °C in a BioTek SYNERGY H1 microplate reader with continuous shaking at 205 cpm. $OD_{600}$ readings were acquired every 10 min. Experiments were performed at least three times with similar results. Results from a representative experiment are shown.

**SOS response activity.** *E. coli* BL21(DE3) were co-transformed with the SOS response reporter plasmid pL(*lexO*)-GFP[56] and the indicated arabinose-inducible effector expression vector. Overnight cultures that were grown in 2xYT media containing the appropriate antibiotics and 0.2% (wt/vol) glucose (to repress expression from Pbad promoter) were washed with fresh 2xYT. Washed suspensions were normalized to $OD_{600}$ = 0.5 in 900 μL of 2xYT supplemented with appropriate antibiotics and either 0.2% (wt/vol) glucose or 0.1% (wt/vol) L-arabinose, to repress or induce gene expression, respectively. Cultures were grown at 37 °C for 2.5 h with agitation (215 rpm), before cells were pelleted and washed with 1 mL of M9 media. Cells were then resuspended in 800 μL of M9 media, and 200 μL of cell suspension was transferred, in triplicate, into wells of a black 96-well plate with a clear optical bottom. GFP fluorescence (excitation 479 nm and emission 520 nm) and $OD_{600}$ were measured using a BioTek SYNERGY H1 microplate reader. Data are presented as arbitrary units resulting from the division of fluorescence measurements by $OD_{600}$ per well. The experiment was repeated three times with similar results. Results from a representative experiment are shown.

**Protein expression in E. coli.** To test the expression of C-terminally myc-tagged V12_14465 variants from pBAD-based L-arabinose-inducible expression vectors in *E. coli*, nontoxic proteins were transformed into *E. coli* BL21(DE3). Toxic variants were co-transformed into *E. coli* BL21(DE3) together with pBAD33.1 containing the immunity gene, *v12_14460*, C-terminally tagged with Flag, to provide protection against self-intoxication and allow expressing cells to grow. Overnight bacterial cultures were diluted 100-fold into 5 mL of fresh 2xYT media supplemented with appropriate antibiotics and grown for 2 h at 37 °C with agitation (215 rpm). Protein expression was induced by addition of L-arabinose (0.1% wt/vol) to the media and an additional incubation at 30 °C for 4 h. Next, 1 mL of each culture was pelleted and the cells were resuspended in 100 μL of (2×) tris-glycine SDS sample buffer followed by boiling at 95 °C for 5 min. Samples were resolved on precast gels (GenScript), transferred onto PVDF membranes, and immunoblotted with c-Myc (9E10) antibodies (sc-40, Santa Cruz Biotechnology) at 1:1000 dilution. Loading of total protein lysates was visualized by Ponceau S straining of the immunoblot membrane. Protein signals were visualized by ECL. Experiments were repeated at least three times with similar results; the results from representative experiments are shown.

**In vitro DNase assay.** Overnight cultures of *E. coli* BL21(DE3) containing plasmids for arabinose-induced expression of the indicated Myc-6xHis-tagged V12_14465 variants and the Flag-tagged V12_14460 (the immunity protein required to allow the toxic V12_14465 variants inside the cells to accumulate) were diluted 100-fold into 100 mL 2xYT media supplemented with kanamycin and chloramphenicol and incubated at 37 °C with agitation (215 rpm). Protein expression was induced by adding 0.1% (wt/vol) L-arabinose when cultures reached an $OD_{600}$ of ~ 1.0, followed by incubation at 30 °C for 4 h with agitation (215 rpm). Cells were harvested by centrifugation and then resuspended in 5 mL lysis buffer A (20 mM Tris-Cl pH 7.5, 500 mM NaCl, 5% vol/vol glycerol, 10 mM immidazole, 1 mM PMSF, and 6 M urea). Urea was used to denature the proteins so that

V12_14460 (the immunity protein; see above) can be released from V12_14465. Cells were disrupted using a high-pressure cell disruptor (Constant system One Shot cell disruptor, model code: MC/AA). Cell debris was removed by centrifugation for 20 min at 13,300 × *g* at 4 °C. Supernatant fractions of lysed cells containing the denatured His-tagged V12_14465 variants were mixed with 100 μL Ni-Sepharose resin (50% slurry; GE healthcare) and incubated for 1 h at 4 °C with constant shaking. The suspensions (cell lysate with Ni-Sepharose resin) were loaded onto a column. Immobilized resin was washed with 10 mL wash buffer A (20 mM Tris-Cl pH 7.5, 500 mM NaCl, 5% vol/vol glycerol, 40 mM imidazole, and 6 M urea). Bound proteins were eluted from the column using 1 mL of elution buffer A (20 mM Tris-Cl pH 7.5, 500 mM NaCl, 5% vol/vol glycerol, 500 mM imidazole, and 6 M urea). Buffers were kept at 4 °C. The presence and purity of the eluted proteins were confirmed by SDS-PAGE.

To refold the denatured, purified proteins before in vitro DNase activity assays, a step-wise refolding procedure was applied. First, the eluted proteins were diluted twofold in ice-cold refolding buffer (20 mM Tris-Cl pH 7.5, 500 mM NaCl, 5% vol/vol glycerol) and incubated for 10 min on ice. This step was repeated, and then the 4 mL suspension was concentrated to 300 μL using a Spin-X UF concentrator column (Corning; 30 kDa or 5 kDa columns were used for full-length or truncated versions of V12_14465, respectively). The concentrated protein suspension was diluted in 3 mL of ice-cold DNase assay buffer (10 mM Tris-Cl pH 7.5, and 50 mM NaCl). The concentration and dilution in DNase assay buffer steps were repeated twice, resulting in 300 μL of concentrated protein in DNase assay buffer. The purified proteins were quantified by the Bradford method using 5X Bradford reagent (Bio-rad).

For in vitro DNase activity, *E. coli* BL21(DE3) isolated genomic DNA (100 ng) was incubated with 1 μg of purified V12_14465 variants (as explained above) for 30 min at 37 °C in DNase assay buffer supplemented with either 1 mM EDTA or 2 mM $MgCl_2$. The reactions were stopped by adding 6.65 μg of Proteinase K and the samples were incubated at room temperature for 10 more minutes. Samples were analyzed on 1.0% agarose-gel electrophoresis. For positive and negative controls, 1 μg DNase I (Bioworld) and 1 μg BSA (Sigma) were used, respectively.

**In vivo DNase assay.** *Escherichia coli* BL21(DE3) containing the indicated pBAD/Myc–His plasmids for expression of BC3021 variants were grown overnight in 2xYT media supplemented with kanamycin and 0.2% (wt/vol) glucose. Overnight cultures were washed with 2xYT and normalized to an $OD_{600}$ of 1.0 in 3 mL of fresh 2xYT supplemented with kanamycin and 0.1% (wt/vol) L-arabinose (to induce protein expression). Cultures were grown for 2 more hours with agitation (215 rpm) at 37 °C before 1.0 $OD_{600}$ units were pelleted. Genomic DNA was isolated from each sample using the EZ spin column genomic DNA kit (Bio Basic) and eluted with 30 μL of sterile distilled water (Milli-Q). Equal genomic DNA elution volumes were analyzed by 0.7% (wt/vol) agarose gel electrophoresis. The experiments were repeated three times with similar results. Results from a representative experiment are shown.

**In vivo RNase assay.** *E. coli* BL21(DE3) containing the indicated pBAD/Myc–His plasmids for expression of BC3021 and V12_14465 variants were grown and induced as described above for the in vivo DNase assay, except that the growth media were supplemented with 0.4% (wt/vol) glucose, and that arabinose induction time was 1 h. Total RNA was purified from 0.5 $OD_{600}$ units of cells using the EZgene™ systems bacterial RNA kit (BIOMIGA) following the manufacturer's instructions. RNA was eluted with 100 μL DEPC-treated water, and 50 μL was further treated with 1 U of DNase I for 30 min. Twenty microliter samples were analyzed by 1% (wt/vol) agarose gel electrophoresis. The experiment was repeated three times with similar results. Results from a representative experiment are shown.

**Fluorescence detection of DNase activity during competition.** For assessment of prey DNA morphology during bacterial competition, the indicated *V. parahaemolyticus* 12-297/B strains were grown overnight and mixed as described above for the bacterial competition assay. The Δ*v12_14465-0* prey harbored a pGFP plasmid (a high copy number plasmid for constitutive expression of GFP[57]) to allow for their distinction from attacker cells. Attacker:prey mixtures were spotted on MLB agar plates and incubated at 30 °C for 3 h. Bacteria were then scraped from the plates, washed with M9 media, and incubated at room temperature for 10 min in M9 media containing the Hoechst 33342 (Invitrogen) DNA dye at a final concentration of 1 μg/μL. The cells were then washed and resuspended in 10 μL of M9 media. One microliter of bacteria suspensions was spotted onto M9 agar (1.5% wt/vol) pads and allowed to dry for 2 min before the pads were placed face-down in sterile 35 mm glass bottom Cellview cell culture dishes. Bacteria were imaged in a Nikon Eclipse Ti2-E inverted motorized microscope equipped with a CFI PLAN apochromat DM 100X oil lambda PH-3 (NA, 1.45) objective lens, Lumencor SOLA SE II 395 light source, and ET-DAPI (#49028, used to visualize Hoechst signal) and ET-EGFP (#49002, used to visualize GFP signal) filter sets. Images were acquired using a DS-QI2 Mono cooled digital microscope camera (16 MP), and were postprocessed using Fiji ImageJ suite[58]. Cells exhibiting the indicated DNA morphologies were manually counted (100 GFP-expressing prey cells per treatment in each experiment). The experiment was repeated three times.

**Pull-down assay**. Overnight cultures of *E. coli* BL21(DE3) containing plasmids for arabinose-induced expression of the indicated *Myc*-6xHis-tagged V12_14465 variants and the Flag-tagged V12_14460 (the immunity protein required to allow accumulation of the toxic V12_14465 variants inside the cells) were diluted 100-fold into 100 mL 2xYT media supplemented with kanamycin and chloramphenicol and incubated at 37 °C with agitation (215 rpm). Protein expression was induced by adding 0.1% (wt/vol) L-arabinose when cultures reached an OD$_{600}$ of ~ 1.0, followed by incubation at 30 °C for 4 h with agitation (215 rpm). Cells were harvested by centrifugation and then resuspended in 5 mL lysis buffer B (20 mM Tris-Cl pH 7.5, 500 mM NaCl, 5% vol/vol glycerol, 10 mM immidazole, and 1 mM PMSF). Cells were disrupted using a high-pressure cell disruptor (Constant system One Shot cell disruptor, model code: MC/AA). Cell debris was removed by centrifugation for 20 min at $13,300 \times g$ at 4 °C. Next, 50 μL of supernatant fractions of lysed cells was taken and mixed with 50 μL of 2× protein sample buffer, boiled at 95 °C for 5 min, and kept for subsequent input protein analysis. Then 200 μL of supernatant fractions of lysed cells (400 μL was used for V12_14465$^{1-282}$ due to its lower expression level compared with other V12_14465 variants) was mixed with 30 μL Ni-Sepharose resin (50% slurry; GE healthcare) and incubated for 1 h at 4 °C with constant shaking. Resin was collected by centrifugation at $2000 \times g$ at 4 °C, and washed with 2.5 mL wash buffer B (20 mM Tris-Cl pH 7.5, 500 mM NaCl, 5% vol/vol glycerol, and 40 mM imidazole). Bound proteins were eluted from the resin using 200 μL of elution buffer B (20 mM Tris-Cl pH 7.5, 500 mM NaCl, 5% vol/vol glycerol, and 500 mM imidazole). Buffers were kept at 4 °C.

The eluted fractions (output) were mixed with 200 μL 2× protein sample buffer and boiled at 95 °C for 5 min. Samples were resolved on TGX stain-free gels (Bio-Rad) and transferred onto PVDF membranes. For V12_14465$^{1-282}$, we loaded twice the volume of input and output samples to equalize the levels of V12_14465 variants. The presence of V12_14465 variants was detected using c-Myc (9E10) antibodies (sc-40, Santa Cruz Biotechnology), and the presence of V12_14460 was detected using Flag antibodies (DYKDDDDK Tag (D6W5B) Rabbit mAb #14793; Cell Signaling Technology). Protein signals were visualized by ECL. The experiment was repeated at least three times with similar results, and the results from a representative experiment are shown.

**BACTH assay**. The adenylate cyclase-based BACTH Gateway system[52] was used. Gateway-compatible plasmids producing the indicated proteins fused to either the T18 or T25 catalytic domains of *Bordetella* adenylate cyclase were transformed into the *E. coli* BTH101 reporter strain. Three independent colonies of each transformant were grown overnight in 2xYT media supplemented with appropriate antibiotics and IPTG (0.5 mM) at 30 °C. The following day, 5 μL of each overnight culture was spotted onto LB agar plates supplemented with ampicillin, spectinomycin, bromo-chloro-indolyl-galactopyrannoside (X-gal, 40 mg/mL), and IPTG (0.5 mM). Plates were incubated for 24 h at 30 °C. The experiment was repeated three times with similar results. Results from a representative experiment are shown.

**TBLASTN search for auxiliary T6SS modules**. TBLASTN was performed against the nucleotide sequence of *V. parahaemolyticus* 12-297/B using the protein sequences of *V. parahaemolyticus* RIMD 2210633-secreted components as queries: NP_797772.1 (Hcp1), NP_800537.1 (Hcp2), NP_797773.1 (VgrG1), NP_800536.1 (VgrG2), NP_797794.1 (amino acids 37-143, PAAR1) and NP_800535.1 (PAAR2). The expect value threshold was set to $10^{-6}$.

**Identification of isolates containing the VgrG1b module**. The nucleotide sequences of *V. parahaemolyticus* isolates were downloaded from NCBI on December 28, 2018, and TBLASTN was performed using the protein sequences of VgrG1b, WP_029857615.1 (V12_14465), and WP_024703575.1 (V12_14460) as queries. The expect value threshold was set to $10^{-6}$. The results were merged and sorted by their nucleotide accession and alignment start position, then manually inspected to identify the VgrG1b modules.

**Identification of FIX- and PoNe-containing proteins**. Position-Specific Scoring Matrices of FIX and PoNe were constructed using amino acids 1–282 and 283–449 of V12_14465, respectively. Five iterations of PSI-BLAST[59] were performed against the reference protein database (a maximum of 500 hits with a threshold of $10^{-6}$ were used in each iteration). Bacterial genomes containing FIX and PoNe were identified and their protein sequences and feature tables were downloaded from NCBI on November 27–29, 2018.

Reverse position-specific BLAST[59] was used to identify the FIX- and PoNe-containing proteins in the bacterial genomes. The results were filtered using an expect value threshold of $10^{-9}$. With PoNe, the protein length had to be at least 80 aa. Unique accessions located at the ends of the contigs were removed. Unique accessions appearing in the same organism in more than one contig were removed if the same downstream gene existed at the same distance.

The proteins were clustered using CD-HIT v4.17[60]. The clustering threshold was set to 40% identity with a minimal 50% length difference (i.e., the shorter protein must be at least 50% of the longer protein), and the sequences were clustered to the most similar cluster that met the threshold. FIX-containing proteins were clustered according to their C-terminal sequences (i.e., sequences

located downstream of FIX), and PoNe-containing proteins were clustered according to their N-terminal sequences (i.e., sequences located upstream of PoNe).

The proteins located upstream of the FIX- and PoNe-containing proteins on the same strand were defined as upstream proteins, whereas proteins located downstream on the same strand were defined as downstream proteins. Protein sequences were analyzed using the NCBI Batch Conserved Domains-Search Tool[61]. Transmembrane topology and signal peptides were predicted using the Phobius server[62]. Signal peptide cleavage sites were predicted using the SignalP 4.1 server[63]. Proteins were considered as containing a signal peptide only if they were designated as such by both Phobius and SignalP 4.1 predictions.

**Construction of the phylogenetic tree**. The DNA sequences of *rpoB* coding for DNA-directed RNA polymerase subunit beta were downloaded from NCBI on December 8, 2018 for bacterial strains encoding PoNe. Partial sequences were removed. The sequences were aligned using MAFFT v7.408 FFT-NS-2[64,65]. The evolutionary history was inferred using the neighbor-joining method[66] with the Jukes–Cantor substitution model (JC69). The analysis involved 1303 nucleotide sequences and 2535 conserved sites. Evolutionary analyses were conducted using the MAFFT server (https://mafft.cbrc.jp/alignment/server/), and the tree was visualized using MEGA 7[67].

**Illustration of conserved residues using WebLogo 3**. FIX- and PoNe-sequences were aligned using MUSCLE[68], and aligned columns not represented in WP_029857615.1 were removed. Conserved residues were illustrated using the WebLogo 3 server (http://weblogo.threeplusone.com[69]). Amino acid numbering was based on the sequence of WP_029857615.1.

**Identification of FIX-containing genomes encoding T6SS**. RPS-BLAST[59] was employed to identify the T6SS core components in the FIX-containing bacterial genomes. The proteins were aligned against 11 COGs (COG3516, COG3517, COG3157, COG3521, COG3522, COG3455, COG3523, COG3518, COG3519, COG3520, and COG3515) shown to specifically predict T6SS and VgrG (COG3501), which was correlated with T6SS-associated organisms[3]. Bacterial genomes encoding at least nine out of the eleven T6SS core components were regarded as harboring T6SS.

**VgrG sequence alignment**. To perform the VgrG sequence alignment, protein sequences of *V. parahaemolyticus* 12-297/B VgrG1 (B5C30_RS15295) and VgrG1b (translated from the nucleotide sequence found between positions 22692 and 20311 in reference sequence NZ_MYFG01000470.1) were aligned using Clustal Omega[70], and the alignment was visualized using JalView[71].

**Secondary-structure prediction**. Secondary structure prediction of the PoNe conserved region of V12_14465 was performed using the Jpred 4 server[72].

**Statistical analysis**. Data were statistically analyzed with GraphPad Prism 6, using the unpaired, two-tailed Student *t* test. Differences were considered significant at $P < 0.05$; n.s. denotes not significant.

**Reporting summary**. Further information on research design is available in the Nature Research Reporting Summary linked to this article.

## Data availability
The experimental and computational data that support the findings of this research are available in this article and its Supplementary Information files, or upon request from the corresponding authors. The source data underlying Figs. 1c–f, 2c–d, 3b, 4a, f, and Supplementary Figs. 2a, b, 3, 4, 5b, c, 7, and 9c are provided as a Source data file.

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

## Acknowledgements

This project received funding from the European Research Council (ERC) under the European Union's Horizon 2020 research and innovation program (Grant agreement no. 714224) (DS). DS is an Alon Fellow. We thank Hanna Engelberg-Kulka and Scot Ouellette for their generous gift of plasmids, Avigdor Eldar for providing bacterial strains, Kinga Keppel and Chen Yacobsohn for excellent technical assistance, and Udi Qimron for his critical reading of the paper. We also thank members of the Salomon lab for fruitful discussions.

## Author contributions

B.J., E.B. and D.S. designed the study. B.J., C.M.F and D.S. performed the experiments. E.B. performed the computational analyses. B.J., C.M.F, E.B. and D.S. analyzed the data. D.S. wrote the paper, and the other authors contributed to the writing. All authors read and approved the final version of the paper.

## Additional information

**Competing interests:** The authors declare no competing interests.

