## [Peer Review File · Nature Communications]

Reviewers' comments:

Reviewer #1 (Remarks to the Author):

The work presented in this manuscript reports on the identification of a yet another toxin/immunity (T/I) pair associated with the type VI secretion system (T6SS). The toxin is encoded within an orphan *vgrG* cluster from *Vibrio parahaemolyticus* and appears to be a DNase with a recognizable catalytic motif. The cognate immunity seems to protect from the antibacterial action of the toxin through direct binding. Most remarkably the authors identified a conserved motif (80 residues near the N terminus), which is found in many other putative T6SS toxins, called FIX. Unfortunately, the role of this motif is not clearly elucidated which would be a very consistent added value to increase the impact of the paper.

Specific comments

- The study identified a novel T6SS toxin with DNase activity. The only readout to monitor T6SS-dependent secretion is killing of sensitive preys. There is no assay showing protein secretion. This should be easy to perform by tagging the toxin appropriately (myc or His as shown in figure 3b) and follow secretion using western immune blot.
- In figure 2f competition against *E. coli* is conducted and one observed that a *hcp1* (likely T6SS1) mutant is no longer killing. The authors previously mentioned the existence of a T6SS2 (line 66). Does this mean that T6SS2 has no antibacterial activity?
- *VgrG1b* has an additional 100 amino acids as compared to *VgrG1* (line 107). Is there any information contained in this extra domain that will be required for recognition of the newly identified toxin? It would be worth trying whether truncation of this domain prevents delivery of the DNase (as tested in figure 1f).
- The work on the immunity showing it protects from the DNase activity and it can interact directly with it (bacterial two hybrid and pull-down) is rather solid. What would be interesting to assess is whether cross protection can actually be provided. For example, can the immunity from the *B. cereus* toxin protect from *V. parahaemolyticus* toxin? Also, what is exactly the difference between immunity with DUF1910 or DUF1911. Could an immunity DUF1911 protect from a toxin linked to an immunity DUF1910 and vice versa?
- The discovery of the FIX motif by this group is an elegant follow up on their previous discovery of a MIX motif. Despite the huge interest one may have with respect to these motifs it is still puzzling what their precise function might be. As the authors indicate that MIX and FIX are mutually exclusive (line 298) it would be important to understand whether they can be exchanged without loss of targeting/secretion of the effector.
- The FIX motif is proposed to be between residues 43-121. There is no clear demonstration that affecting only this region will impact the toxin delivery. The assay using truncated form of the toxin (Figure 4a) uses various truncations (for example starting at position 95) but for example does not show a version from which 20 or 30 residues from within the FIX motif have been removed. This will be more compelling evidence. Also, there is no indication that any of the truncated forms do fold properly or are stable. This needs to be tested either by western blot (stability) or DNase assay (activity). If the protein is no longer active then there will be no killing of the prey in any case. One option is also to monitor secretion using the FIX truncated forms.
- As said the role of FIX is not clearly discussed. Since FIX can be found in proteins carrying PAAR or *VgrG* (figure 4c) this suggests that this cannot be the only motif targeting or associating the toxin with the machinery. It would be important to figure out which component in the machine FIX may interact with, if at all. One possible experiment could be to produce the FIX domain only and perform crosslinking experiments to check with which component it interacts with. Alternatively, photo-crosslinking experiment using photoactivable residues introduced within the FIX domain might also be doable to see with which components it interacts.

Minor comments

- The term "marker" for T6SS substrates as indicated in the title is very vague and does not say much about its putative role other than being found exclusively in this category of proteins. This

definitely falls short of making any high impact.

- It is indicated that mostly antibacterial activity of T6SS effectors have been identified (line 49). However, there is still a long list of anti-eukaryotic T6SS effectors as reported in a review by Hachani et al (Current Opinion in Microbiology, 2016).
- Line 55, it could be useful to specify that the conserved protein is FtsZ and that is involved in cell division.
- Figure 2b is not needed.
- Panels d and e from figure 2 should actually be the other way around according to the figure legend.
- It is indicated that exogenous overexpression of the toxin is toxic (lines 130-131). Is there a figure showing this when the toxin is overexpressed in *E. coli*? Should this refer to supplementary figure 4?
- Overall it would be useful for the reader that a name is given to the new DNase toxin and its cognate immunity and be kept all through (rather than the V number). Why not using PoNevp and PoNivp, as for example in figure 3b.
- The classification presented in figure 2c, showing the PoNe connection with cognate secretion system is slightly confusing. One would expect that a protein transported by the T5SS has a signal peptide. Yet it does not seem to be the case. Curiously those that have a signal peptide are linked to a T2SS. Why will it not be a T5SS? This would need to be seriously reviewed to avoid spreading confusion. Also, what is the basis for the identification of the signal peptide? Sometime there are wrong annotations using the wrong start codon and therefore signal peptide could be missed. The authors are also not confident about the T2SS connection since they do not indicate this secretion system as a possible one on line 270.
- In figure 3 legend there is no indication on what OpaR is. Instead this is clearly mentioned in legend of supplementary figure 4. Could have been the other way around.

Reviewer #2 (Remarks to the Author):

The type VI secretion system (T6SS) is a widespread multi-protein assembly used by Gram-negative bacteria to antagonize competitors. Attacking cells directly inject toxic proteins called effectors into recipient cells during a T6SS attack. Effector proteins have a wide variety of targets in recipient cells and characterized effectors typically target essential pathways or molecules. Here, Jana and Fridman et al. characterize a novel DNA-targeting toxin, PoNe, encoded within the *Vibrio parahaemolyticus* genome. The group starts by characterizing the T6SS-dependence of this toxin followed by characterizing its cognate immunity protein, PoNi. In addition to this, they have found that the N-terminal portion of this effector contains a "FIX" domain, which is necessary for delivery of the toxin into recipient cells but does not affect toxicity of the effector. Overall, the group identifies a novel T6SS toxin family and uncover an unconventional mode of trafficking and recognition prior to delivery. Though other T6SS DNase effectors have been described in the literature, this work is much more thorough compared to these prior reports. The manuscript is well written, and experiments are nicely controlled. I have a couple of points that I would like to the authors to address.

Minor comments:

- Line 35 and 36, add "the" before "type VI secretion system".
- Line 55, NAD⁺ and NAD(P)⁺ but not their reduced forms (NAD(P)H) have been shown to be targets of Tse6.

Major comments:

- Does the nuclease activity for PoNe extend to RNA? If so, what types of RNA can it target (tRNA, rRNA)? I would like the authors to test if PoNe enzymes target ribonucleic acids in addition to deoxyribonucleic acids.

- Can a homology model for this protein be confidently predicted? If so, this will help readers visualize the identified catalytic motif.
- PoNe activity is shown in vitro and artificially in *E. coli* but it would be nice to observe this during interbacterial competition with *Vibrio* cells delivering this effector into susceptible competitors. DAPI staining on cells during interbacterial competition would accomplish this goal.
- My concern with the FIX domain (and by extension, the previously published MIX domain) is that it will not be widely used as a marker for effectors by the T6SS field. The discovery of these domains is very interesting and opens up many avenues for investigation but I worry that because these domains are defined by an HMM displayed in a figure rather than in protein domain family databases (i.e. PFAM, InterPro, etc.), their significance will be lost. Is it possible for the authors to communicate with the curators of these databases so that these domains can be defined and thus be more accessible to other researchers?
- Using an approach similar to either Fig. 3c or 3d, I would like the authors to test if FIX domains physically interact with VgrGs. The very nice informatics work performed by these authors identified examples of FIX domains fused to VgrG and PAAR domains, suggesting they likely interact with these structural components.

Reply to reviewers' comments:

Reviewer #1 (Remarks to the Author):

The work presented in this manuscript reports on the identification of a yet another toxin/immunity (T/I) pair associated with the type VI secretion system (T6SS). The toxin is encoded within an orphan vgrG cluster from *Vibrio parahaemolyticus* and appears to be a DNase with a recognizable catalytic motif. The cognate immunity seems to protect from the antibacterial action of the toxin through direct binding. Most remarkably the authors identified a conserved motif (80 residues near the N terminus), which is found in many other putative T6SS toxins, called FIX. Unfortunately, the role of this motif is not clearly elucidated which would be a very consistent added value to increase the impact of the paper.

Specific comments

1) The study identified a novel T6SS toxin with DNase activity. The only readout to monitor T6SS-dependent secretion is killing of sensitive preys. There is no assay showing protein secretion. This should be easy to perform by tagging the toxin appropriately (myc or His as shown in figure 3b) and follow secretion using western immune blot.

We have made various attempts to directly visualize the secretion of the effector using immunoblot techniques, but have not been able to detect a signal in the supernatant fractions. We have witnessed this happen also with other effectors which we assume are secreted at very low amounts and whose secretion may require exact stoichiometric ratios of other components of the secretion system. The use of T6SS-dependent toxicity against sensitive prey has been used in numerous papers and is widely accepted as a reliable indication of T6SS-mediated delivery of effectors. Taken together with the presence of the effector in a VgrG module, and its dependency on that VgrG for delivery, we are confident that we found a bona fide T6SS effector.

2) In figure 2f competition against *E. coli* is conducted and one observed that a hcp1 (likely T6SS1) mutant is no longer killing. The authors previously mentioned the existence of a T6SS2 (line 66). Does this mean that T6SS2 has no antibacterial activity?

Indeed, all *V. parahaemolyticus* isolates harbor T6SS2. We have previously shown that T6SS2 is not active under the warm marine-like conditions used here to activate T6SS1 (i.e. 30°C, 3% NaCl, and in the presence of surface sensing activation provided by the agar plate on which the competition occurs) (Salomon et al., PLoS One, 2013) and therefore does not interfere with assessment of T6SS1 antibacterial activity. We also added an explanation of the hcp1 deletion strain in the legend of Fig. 1.

3) VgrG1b has an additional 100 amino acids as compared to VgrG1 (line 107). Is there any information contained in this extra domain that will be required for recognition of the newly identified toxin? It would be worth trying whether truncation of this domain prevents delivery of the DNase (as tested in figure 1f).

We now show that the C-terminal 101 residues of VgrG1b are required for proper delivery of the effector into the prey cells (Fig. 1e, supplementary Fig 4, and lines 144-161). We also demonstrate that while these 101 residues are not sufficient to support effector delivery when fused to the T6SS1 cluster-encoded VgrG1, replacing residues 520-692 of VgrG1 with residues 520-793 of VgrG1b results in a fusion protein that can partially support effector delivery. These new data suggest that information encoded within the VgrG1b C-terminus is mediating effector delivery into target cells.

4) The work on the immunity showing it protects from the DNase activity and it can interact directly with it (bacterial two hybrid and pull-down) is rather solid. What would be interesting to assess is whether cross protection can actually be provided. For example, can the immunity from the *B. cereus* toxin protect from *V. parahaemolyticus* toxin?

We now show that PoNi provide immunity specifically against their adjacently-encoded PoNe, and do not cross-protect against PoNe from another bacterium (see Supplementary Fig. 9e, lines 246-249)

Also, what is exactly the difference between immunity with DUF1910 or DUF1911. Could an immunity DUF1911 protect from a toxin linked to an immunity DUF1910 and vice versa?

In most cases, both DUF1911 and DUF1910 domains are present together in the immunity protein (see Supplementary Dataset 1). Yet in some immunity proteins, like V12_14460, only DUF1911 was predicted without the accompanying DUF1910. We have now modified the text to clarify this point (lines 224-230).

5) The discovery of the FIX motif by this group is an elegant follow up on their previous discovery of a MIX motif. Despite the huge interest one may have with respect to these motifs it is still puzzling what their precise function might be. As the authors indicate that MIX and FIX are mutually exclusive (line 298) it would be important to understand whether they can be exchanged without loss of targeting/secretion of the effector.

We do not yet know the exact role of either MIX or FIX in secretion or delivery of T6SS effectors, and this topic is under ongoing investigation. While we show here that FIX is required for T6SS-mediated delivery of the effector, we have yet to determine whether FIX (or MIX) is also sufficient for T6SS-mediated delivery or not. Therefore, while we agree

that the proposed experiment is of great interest, we are unable at this time to provide additional data to support a claim for the specific role of FIX. Our current data allow us to conclude that FIX is required for proper delivery of V12_14465 from the attacker cell to the cytoplasm of the recipient cell where it is expected to work (see reply to comment #6 below), and to propose its use as a marker to identify novel T6SS substrates. We hope to be able to provide a better understanding of the mechanistic role and function of FIX (and MIX) in future work.

6) The FIX motif is proposed to be between residues 43-121. There is no clear demonstration that affecting only this region will impact the toxin delivery. The assay using truncated form of the toxin (Figure 4a) uses various truncation (for example starting at position 95) but for example does not show a version from which 20 or 30 residues from within the FIX motif has been removed. This will be more compelling evidence.

As suggested by the reviewer, we now show that deletion of an inner part of FIX, i.e. amino acids 70-89 containing several conserved residues, results in a mutant that is still toxic when expressed in the cytoplasm of E. coli but is no longer functional in a competition assays (see Fig. 4f, Supplementary Fig. 5, and lines 286-290).

Also, there is no indication that any of the truncated form do fold properly or is stable. This needs to be tested either by western blot (stability) or DNase assay (folding). If the protein is no longer active then there will be no killing of the prey in any case. One option is also to monitor secretion using the FIX truncated forms.

Expression (immunoblots) and folding (toxicity in E. coli) of all V. parahaemolyticus V12_14465 effector forms tested in this work are presented in Supplementary Fig. 5.

7) As said the role of FIX is not clearly discussed. Since FIX can be found in proteins carrying PAAR or VgrG (figure 4c) this suggests that this cannot be the only motif targeting or associating the toxin with the machinery. It would be important to figure out which component in the machine FIX may interact with, if at all. One possible experiment could be to produce the FIX domain only and perform crosslinking experiments to check with which component it interacts with. Alternatively, photo-crosslinking experiment using photoactivable residues introduced within the FIX domain might also be doable to see with which components it interacts.

The newly added results showing that the C-terminus of VgrG1b is required and sufficient for delivery of the effector into prey cells (Fig. 1e) strongly support the notion that the effector interacts with its upstream VgrG1b. The new results obtained with the internal FIX deletion (of residues 70-89) support the notion that FIX is required for proper delivery of the effector from one cell to another (Fig. 4f). Nevertheless, while we have made many attempts to determine if the effector, either through FIX or not, directly binds

VgrG1b using bacterial two-hybrid, pull-down, and co-IP assays, we have not been able to obtain conclusive evidence to support or dismiss it. Therefore, at this time we cannot say with certainty whether FIX is directly mediating interaction with the T6SS tube. We are considering other roles of FIX as well, but these hypotheses are at preliminary stages of investigation and are beyond scope of the current manuscript (please also see reply to comment #5).

Minor comments

8) The term “marker” for T6SS substrates as indicated in the title is very vague and does not say much about its putative role other than being found exclusively in this category of proteins. This definitely falls short of making any high impact.

As explained in our replies to comments #5 and #7, we are not comfortable with stating a general role for FIX beyond its use as a marker for detection of novel T6SS effectors and toxin domains. We do believe that even without determining a specific mechanistic role for FIX, its discovery will have a great impact and interest in the T6SS field (as acknowledged by the reviewer in comment #5), being merely the second domain that is found specifically in T6SS substrates and that allows identification of T6SS-specific effectors.

9) It is indicated that mostly antibacterial activity of T6SS effectors have been identified (line 49). However, there is still a long list of anti-eukaryotic T6SS effectors as reported in a review by Hachani et al (Current Opinion in Microbiology, 2016).

The suggested citation was added (line 53).

10) Line 55, it could be useful to specify that the conserved protein is FtsZ and that is involved in cell division.

The suggested comment was added (line 51).

11) Figure 2b is not needed.

We moved the original Fig. 2b into the supplementary material (now Supplementary Fig. 6).

12) Panels d and e from figure 2 should actually be the other way around according to the figure legend.

We apologize for this oversight. This issue had been resolved in the revised Fig. 2.

13) It is indicated that exogenous overexpression of the toxin is toxic (lines 130-131). Is there a figure showing this when the toxin is overexpressed in E. coli? Should this refer to supplementary figure 4?

Yes, this referred to Supplementary Fig. S4, which is now Supplementary Fig. 5. The relevant Supplementary Figure is cited in the text (line 169) and in the legend of Fig. 2.

14) Overall it would be useful for the reader that a name is given to the new DNase toxin and its cognate immunity and be kept all through (rather than the V number). Why not using PoNevp and PoNivp, as for example in figure 3b.

We have tried naming the effector as PoNe^{Vp}, but we found that it becomes confusing and it is unclear whether the domain or the entire effector is discussed. Since we are mostly describing a single effector, V12_14465, we think that the readers will not be confused by the name. To simplify the situation for the readers, we have changed all text and figures to denote the effector as V12_14465 (with its different derivatives as following superscript) and the toxin domain as PoNe.

15) The classification presented in figure 2c, showing the PoNe connection with cognate secretion system is slightly confusing. One would expect that a protein transported by the T5SS has a signal peptide. Yet it does not seem to be the case. Curiously those that have a signal peptide are linked to a T2SS. Why will it not be a T5SS? This would need to be seriously reviewed to avoid spreading confusion. Also, what is the basis for the identification of the signal peptide? Sometime there are wrong annotations using the wrong start codon and therefore signal peptide could be missed. The authors are also not confident about the T2SS connection since they do not indicate this secretion system as a possible one on line 270.

As the reviewer suggested, we have amended Fig. 2b (previously Fig. 2c). Predicted secretion systems are noted based on domains that are established as those found in effectors of the indicated system, without regard to the presence or absence of signal peptide. Also, to prevent confusion, we changed the title of the relevant column from "Predicted secretion system" to "Predicted association with secretion system". This is meant to clarify that we predict association of PoNe domains with various secretion systems is based on domains that are known to be associated with such systems.

As for the prediction of a signal peptide, we note in the Methods section (lines 691-694) that we considered a protein sequence to contain a signal peptide only if one was positively predicted by both SignalP 4.1 and Phobius prediction servers. We acknowledge that T5SS effectors are expected to possess a signal peptide, and indeed some are predicted to have one. Several others have an N-terminal domain named ESPR (Extended Signal Peptide of Type V Secretion System) according to the NCBI Conserved Domains database, but the E Values of these predictions are below our strict confidence cutoff (as mentioned in the Methods section) and were thus not listed.

16) In figure 3 legend there is no indication on what OpaR is. Instead this is clearly mentioned in legend of supplementary figure 4. Could have been the other way around.

We have added the description on OpaR to the legend of Fig. 3.

Reviewer #2 (Remarks to the Author):

The type VI secretion system (T6SS) is a widespread multi-protein assembly used by Gram-negative bacteria to antagonize competitors. Attacking cells directly inject toxic proteins called effectors into recipient cells during a T6SS attack. Effector proteins have a wide variety of targets in recipient cells and characterized effectors typically target essential pathways or molecules. Here, Jana and Fridman et al. characterize a novel DNA-targeting toxin, PoNe, encoded within the *Vibrio parahaemolyticus* genome. The group starts by characterizing the T6SS-dependence of this toxin followed by characterizing its cognate immunity protein, PoNi. In addition to this, they have found that the N-terminal portion of this effector contains a "FIX" domain, which is necessary for delivery of the toxin into recipient cells but does not affect toxicity of the effector. Overall, the group identifies a novel T6SS toxin family and uncover an unconventional mode of trafficking and recognition prior to delivery. Though other T6SS DNase effectors have been described in the literature, this work is much more thorough compared to these prior reports. The manuscript is well written, and experiments are nicely controlled. I have a couple of points that I would like to the authors to address.

Minor comments:

1) Line 35 and 36, add "the" before "type VI secretion system".

Added.

2) Line 55, NAD⁺ and NAD(P)⁺ but not their reduced forms (NAD(P)H) have been shown to be targets of Tse6.

We thank the reviewer for pointing this out. We have corrected the relevant text.

Major comments:

3) Does the nuclease activity for PoNe extend to RNA? If so, what types of RNA can it target (tRNA, rRNA)? I would like the authors to test if PoNe enzymes target ribonucleic acids in addition to deoxyribonucleic acids.

We examined total RNA purified from *E. coli* cells expressing PoNe, and could see no major difference in the total RNA pattern compared with cells not expressing PoNe or expressing a catalytically inactive mutant (see new Supplementary Fig. 8, lines 200-203). Samples of the same induced bacterial cultures were also simultaneously tested for genomic DNA degradation which was apparent, indicating that the PoNe toxin was

expressed and active against DNA in the time frame of the assay. Moreover, as explained in reply to the reviewer's comment #5, we could see that the DNA of sensitive recipient cells completely disappears in a PoNe-dependent manner during competition, indicating that PoNe acts as a DNase during bacterial competitions. Taken together, these results indicate that PoNe mainly acts as a DNase. While one cannot rule out activity against a specific tRNA or mRNA molecule, it appears that this possibility is unlikely.

4) Can a homology model for this protein be confidently predicted? If so, this will help readers visualize the identified catalytic motif.

We were not able to obtain a high confidence model of the effector using Phyre, CPHmodels, or HHpred with MODELLER. This is in agreement with the notion that protein families belonging to the PD-(E/D)xK superfamily (like PoNe) are highly divergent in both amino acid sequences and structure.

5) PoNe activity is shown in vitro and artificially in E. coli but it would be nice to observe this during interbacterial competition with Vibrio cells delivering this effector into susceptible competitors. DAPI staining on cells during interbacterial competition would accomplish this goal.

As suggested by the reviewer, we now show that during competition the DNA of PoNe-susceptible cells disappears (as determined using Hoescht dye) in a manner dependent on the presence of the PoNe effector in the attacker strain (see new Fig. 2d-e, lines 204-212). This results strongly supports our conclusion that PoNe acts as a DNase toxin.

6) My concern with the FIX domain (and by extension, the previously published MIX domain) is that it will not be widely used as a marker for effectors by the T6SS field. The discovery of these domains is very interesting and opens up many avenues for investigation but I worry that because these domains are defined by an HMM displayed in a figure rather than in protein domain family databases (i.e. PFAM, InterPro, etc.), their significance will be lost. Is it possible for the authors to communicate with the curators of these databases so that these domains can be defined and thus be more accessible to other researchers?

We thank the reviewer for this comment. We are in contact with the CDD curation team at NCBI and we expect that FIX, PoNE, and MIX will be added to the CDD database soon.

7) Using an approach similar to either Fig. 3c or 3d, I would like the authors to test if FIX domains physically interact with VgrGs. The very nice informatics work performed by these authors identified examples of FIX domains fused to VgrG and PAAR domains, suggesting they likely interact with these structural components.

The newly added results showing that the C-terminus of VgrG1b is required and sufficient for delivery of the effector into prey cells (Fig. 1e) strongly support the notion that the effector interacts with its upstream VgrG1b. The new results obtained with the internal FIX deletion (of residues 70-89) support the notion that FIX is required for proper delivery of the effector from one cell to another (Fig. 4f). Nevertheless, while we have made many attempts to determine if the effector, either through FIX or not, directly binds VgrG1b using bacterial two-hybrid, pull-down, and co-IP assays, we have not been able to obtain conclusive evidence to support or dismiss it. Therefore, at this time we cannot say with certainty whether FIX is directly mediating interaction with the T6SS tube. We are considering other roles of FIX as well, but these hypotheses are at preliminary stages of investigation and are beyond scope of the current manuscript.

REVIEWERS' COMMENTS:

Reviewer #1 (Remarks to the Author):

This is a manuscript that I reviewed earlier.

The authors overall have done a very good job at addressing several of my comments and mostly acknowledge that further characterizing how FIX may work is beyond the scope of the present paper which I can agree with.

I may still point at two minor issues.

- I do agree that previous work showed that T6SS2 is not expressed in the conditions used but low expression might still be sufficient to assemble a system and deliver a few toxins with inhibitory activity. If a T6SS2 mutant was available, it would not take much to assess whether it is or not still able to challenge *E. coli*.
- Regarding the signal peptide of T5SS substrate it is true that they can be unusually long. If there is a prediction that points into that direction but is below confidence cut-off one may still mention the observation and thus quote as "possible/putative" ESPR.

Reviewer #2 (Remarks to the Author):

The authors have satisfactorily addressed my comments in their revised manuscript.

Response to reviewers' comments:

Reviewer #1 (Remarks to the Author):

This is a manuscript that I reviewed earlier.

The authors overall have done a very good job at addressing several of my comments and mostly acknowledge that further characterizing how FIX may work is beyond the scope of the present paper which I can agree with.

We thank the reviewer for the helpful suggestions.

I may still point at two minor issues.

- I do agree that previous work showed that T6SS2 is not expressed in the conditions used but low expression might still be sufficient to assemble a system and deliver a few toxins with inhibitory activity. If a T6SS2 mutant was available, it would not take much to assess whether it is or not still able to challenge *E. coli*.

Our data clearly demonstrated that the investigated effector is a substrate of T6SS1 as deletion of *hcp1* (to inactivate T6SS1) abolished all effector-mediated toxicity. Therefore, while some basal activity of T6SS2 could contribute to interbacterial toxicity, it does not appear to be relevant to the studied effector.

- Regarding the signal peptide of T5SS substrate it is true that they can be unusually long. If there is a prediction that points into that direction but is below confidence cut-off one may still mention the observation and thus quote as "possible/putative" ESPR.

Since the proposed T5SS substrate have additional T5SS-associated domains besides the putative ESPR in question (that associate these proteins with the T5SS), we prefer not to make exceptions to our confidence cut-offs, and to allow readers to individually test and assess a specific protein of interest.

Reviewer #2 (Remarks to the Author):

The authors have satisfactorily addressed my comments in their revised manuscript.

We thank the reviewer for the helpful suggestions.